# Contraceptive use by number of living children in Ghana: Evidence from the 2017 maternal health survey

Sarah Compton[1]*, Emmanuel Nakua[2], Cheryl Moyer[3], Veronica Dzomeku[4], Emily Treleaven[5], Easmon Otupiri[6], Jody Lori[7]

1 Department of Obstetrics and Gynecology, Program on Women's Healthcare Effectiveness Research, University of Michigan, Ann Arbor, Michigan, United States of America, 2 Department of Epidemiology and Biostatistics, School of Public Health, Kwame Nkrumah University of Science and Technology, Kumasi, Ghana, 3 Department of Learning Health Sciences and Department of Obstetrics and Gynecology, University of Michigan, Ann Arbor, Michigan, United States of America, 4 Department of Nursing and Midwifery, College of Health Sciences, Kwame Nkrumah University of Science and Technology, Kumasi, Ghana, 5 Institute for Social Research, Survey Research Center, University of Michigan, Ann Arbor, Michigan, United States of America, 6 Department of Population, Family, and Reproductive Health, Kwame Nkrumah University for Science and Technology, Kumasi, Ghana, 7 Global Affairs and Community Engagement, School of Nursing, University of Michigan, Ann Arbor, Michigan, United States of America

* sarahrom@med.umich.edu

**Data Availability Statement:** The data underlying the results presented in the study are available from https://www.dhsprogram.com/data/dataset/Ghana_Special_2017.cfm?flag=0.

## Abstract

### Background

There is a significant literature describing the link between parity and contraceptive use. However, there is limited knowledge about the disaggregation by parity of the type of contraceptives. In this study, we describe the use of contraceptives by parity among women of reproductive age in Ghana, focusing on use of highly effective methods (injection, pill, intrauterine device, implant, and sterilization).

### Methods

Using the 2017 Ghana Maternal Health Survey, a nationally-representative cross-sectional household survey, we describe contraceptive method use by number of living children among sexually active women of reproductive age. We then estimated predictors of use of highly effective contraception in a multilevel logistic regression model.

### Results

Most women in this survey are not using any method of contraception, although this varies by whether or not they have begun childbearing. Contraceptive method use varies by number of living children. Before having children, natural (periodic abstinence and withdrawal) and episodic (condoms) methods dominate. Once a woman has one living child, method preference changes to injectables and implants. Factors associated with using a highly effective method of contraception are: having >3 children, being in a relationship, having had an abortion, being younger than age 30, and having had sexual intercourse within days of answering the survey (p < .001 for all).

**Funding:** The author(s) received no specific funding for this work.

**Competing interests:** The authors have declared that no competing interests exist.

## Conclusion

In this analysis, the number of living children a woman has, her age, and timing of last intercourse are the most significant predictors of using a highly effective method of contraception. However, the majority of participants in this study report not using any method of contraception to avoid unwanted pregnancies. Future research that attempts to unpack the disconnect between not wanting to become pregnant and not using contraception is warranted.

## Background

Globally, many countries have experienced a significant decline in their fertility rates [1]. The decline in fertility has been slower across sub-Saharan African (SSA) countries compared to other regions of the world [2]. In Ghana, a small country in West Africa, studies have shown that not all women who desire to use family planning are able to freely access a desired method, while others have shown a stall in fertility decline [3,4]. While use of contraception among Ghanaian women has increased since the mid-1980s, the rate has stalled at less than 25%. In 2006, the Maputo Programme of Action (MPoA), signed by 48 African heads of state, highlighted a need for universal accessibility to sexual and reproductive health (SRH) services [5]. The revised MPoA 2016–2030 provides a renewed and more robust commitment by African leaders to facilitate and provide the enabling environment for attainment of universal access to SRH services. Similarly, in 2012, 41 countries, including Ghana, ratified actions to ensure access to contraceptives [6]. Following these policies and programmes of action, the United Nations and its partners introduced the 17 Sustainable Development Goals (SDGs), of which Target 3.7 seeks to ensure universal access to SRH services, including family planning and contraceptives by the year 2030 [7]. Despite these actions, the use of contraceptives—particularly highly effective methods (e.g., oral contraceptives, injectable contraceptives, intrauterine devices, and implants)—remains low in SSA countries compared to other regions of the world.

Of the estimated 1.1 billion women of reproductive age globally who are currently sexually active, 24.5% report not wanting to become pregnant any time soon, indicating an unmet need for contraception [8]. Although low contraceptive use is a public health concern across the globe, it appears to be an endemic issue in SSA. A study of 17 SSA countries revealed that only 17% of reproductive-aged women use contraceptives [9]. In Ghana, the 2014 Demographic and Health Survey found modern contraceptive use to be low among women aged 15–19 (19%) and 45–49 (18%) [10]. However, in that same survey, more than 40% of respondents reported they do not want to have any more children [11].

A combination of socioeconomic, demographic, cultural, and contextual factors have been found to significantly predict the use of contraceptives. For example, socioeconomic factors such as household wealth, place of residence, and educational attainment have been identified as significant factors that influence women's use of contraceptives [11,12]. Likewise, contextual and cultural issues such as misperceptions about contraceptives, fear of side effects, and level of community literacy play a critical role in influencing women's use of contraceptives [11,13]. From a demographic perspective, there is compelling evidence to show that women's age and parity are significantly associated with their use and type of contraceptives, including the use of highly effective methods [14,15]. Behrman et al. report that in West Africa, nulliparous women had higher modern and highly effective contraceptive use than parous women.

However, in many contexts in SSA, including Ghana, contraceptive use is considered inappropriate until after a first birth because women are under pressure to have children soon after marriage [16]. In the case of Ghana, this is evident in a total fertility rate of 3.9 and a median age at first birth of 21.5 years [17].

Despite evidence of a significant association between parity and contraceptive use, there is limited knowledge in the published literature about the disaggregation by parity of the type of contraceptives used. Thus, the question that remains is, "What is the distribution of contraceptive use by parity in Ghana?" This question remains unanswered in previous studies conducted in Ghana or other SSA countries [18,19], reflecting a significant gap in the current understanding of parity in relation to women's use of contraceptives. The present study seeks to narrow this knowledge gap by investigating the use of contraceptives by parity among women of reproductive age. The analysis disaggregates the various types of contraceptives and the use of highly effective vs. less effective methods by the number of living children the woman has to provide a deeper understanding of the dynamics at play.

## Methods

Using the 2017 Ghana Maternal Health Survey (GMHS) [20], we assessed method of contraception used by the number of living children among respondents who reported they have ever had sexual intercourse. GMHS is a nationally representative cross-sectional household survey with a two-stage stratified random sample. Households were randomly sampled within each enumeration area that was selected as a primary sampling unit with probability proportionate to size. Within each selected household, all women of reproductive age (15–49 years) were eligible to participate. Data collection was implemented by the Ghana Statistical Service and the Ghana Health Service from June 15-October 12, 2017 and analysis was carried out from January-June 2023. All data were anonymized by the Ghana Health Service and Ghana Statistical Service. The authors did not have access to information that could identify participants before or after data collection. The University of Michigan Institutional Review Board deemed this study Not Regulated (HUM00236811). Consent was not required, as only publicly available deidentified data were used.

Age was originally collected continuously in years. We created six categories (<18, 18–24, 25–29, 30–34, 35–39, 40+). Similarly, age at first intercourse was categorized into five groups (<15, 15–16, 17–18, 19–20, 21+). Next, we included a categorical measure of time since last intercourse. Respondents reported whether their last intercourse was "days ago," "weeks ago," "months ago," and "years ago"; responses are categorized as "within the last week," "within 1–4 weeks," "within 1–12 months," or "more than one year." Lastly, we created a dichotomous variable indicating whether a woman is currently using "highly effective contraception," defined as intrauterine device (IUD), implant, injection, pill, or sterilization.

Descriptive statistics are reported with GMHS-defined survey weights. Via logistic regression with standard errors clustered by enumeration area, we analyzed factors associated with using a highly effective method of contraception among women who have had sex (n = 21,397). Those factors associated with the outcome in bivariate Chi-square analysis, and those we felt it important to control for a priori, were included in the multivariate analysis. Covariates included whether a woman is currently in a relationship (either cohabitating or married); educational attainment (none, any primary, any junior secondary, any secondary, any post-secondary/higher); household wealth quintile; place of residence (urban, rural); whether the respondent has ever been pregnant; whether the respondent has ever had an abortion; number of living children; age; age at first intercourse; and time since last intercourse. After running the regression, we conducted the Homer-Lemeshow test to assess the goodness-

of-fit using the estat gof command. The Hosmer-Lemeshow Chi-squared test had a p-value of .458, suggesting no evidence of lack of fit.

We built a second logistic regression model for those who are using any form of contraception (n = 5,968) to investigate, among this group of women, factors associated with using highly effective methods vs. less effective methods. This model includes the same set of covariates listed above, as well as standard errors clustered by enumeration area. The Hosmer-Lemeshow Chi-squared test had a p-value of .222, also suggesting no evidence of lack of fit.

Results are presented as odds ratios and marginal effects (generated using the *margins* command in Stata 16).

## Results

A total of 25,062 women were included in the survey, and 21,397 in the two regression models. The majority of respondents (n = 19,073, 76.1% full sample; n = 15,429, 72.1% regression sample) were not using contraception. Table 1 displays the general demographics of our sample.

For those women who reported using contraception, method choice varied by the number of living children (Fig 1). Those with no living children rely most heavily on periodic abstinence and condoms (23.1% of all women with no children who use contraception use periodic abstinence while 22.1% use condoms), and the pill (14.4%). Injectables became the method of choice for women once they have one living child, although implants increased as a method of choice once a woman had 5 living children, and was the most popular method for those with 6 or more living children.

The results show that 28.5% of women who have been pregnant are currently using contraception, while only 11.2% of those who have not been pregnant are currently using contraception (p < .001) (Fig 2).

In our regression analysis reflecting use of highly effective contraception, participants in a relationship are 4.3% less likely (marginal effect: -.043; p < .001) to be using highly effective contraception, while those who have had an abortion are 4.5% more likely. Compared with people with a primary education, those with no education are 4.6% less likely to use highly effective contraception and higher education is associated with an increased likelihood of using highly effective contraception (3.1%, 2.5%, and 3.3% more likely for those who have attended Junior Secondary, Senior Secondary, and higher, respectively).

Of interest, those with 0, 1, or 2 children, compared to those with 3, are less likely to be using highly effective contraception (19.7%, 12.0%, and 5.6%, respectively), while those with 4, 5, and 6 + living children are more likely to be using highly effective contraception (4.5%, 8.3%, and 11.4%, respectively). Younger participants are more likely to be using highly effective contraception, compared to those aged 30–34. Participants older than age 35 are less likely to be using highly effective contraception. Participants who have had intercourse within days of the survey are 4.5% more likely to be using highly effective contraception than those who have not had intercourse in "weeks," and those who have not had sex in months or years are 8.2% and 15.6% less likely, respectively.

The full results of our logistic regression can be seen in Table 2.

For the second regression model, among those using contraception, the factors associated with using a highly effective method are lower education (those with no formal education are 4.4% more likely than those with primary education to be using highly effective contraception); being in a lower wealth quintile; living in a rural area; having more living children; and having had sex within the last week. The factors negatively associated with using highly effective contraception are older age (those older than age 40 are 7.2% less likely than those aged 30–34 to be using highly effective contraception); higher education; being in a higher wealth quintile; and having no living children. Full results from this model can be seen in Table 3.

**Table 1. Demographics of the study sample.**

| Variable | 0 children N = 8,209 | 1+ children N = 16,853 | Full Sample N = 25,062 | Regression Sample N = 21,397 | P-value[a] |
|---|---|---|---|---|---|
| **Age, years** | | | | | < .001 |
| <18 | 3,863 (47.1) | 300 (1.8) | 4,163 (16.6) | 1,348 (6.3) | |
| 18–24 | 2,957 (36.0) | 3,040 (18.0) | 5,997 (23.9) | 5,236 (24.5) | |
| 25–29 | 778 (9.5) | 3,335 (19.8) | 4,113 (16.4) | 4,043 (18.9) | |
| 30–34 | 289 (3.5) | 3,270 (19.4) | 3,559 (14.2) | 3,545 (16.6) | |
| 35–39 | 159 (1.9) | 2,927 (17.4) | 3,086 (12.3) | 3,083 (14.4) | |
| 40+ | 163 (2.0) | 3,981 (23.6) | 4,144 (16.5) | 4,142 (19.4) | |
| **Education** | | | | | < .001 |
| None | 521 (6.4) | 5,987 (35.5) | 6,508 (26.0) | 6,376 (29.8) | |
| Primary | 1,112 (13.5) | 3,634 (21.6) | 4,746 (18.9) | 4,116 (19.2) | |
| Junior Secondary | 3,219 (39.2) | 4,627 (27.5) | 7,846 (31.3) | 6,072 (28.4) | |
| Senior Secondary | 2,465 (30.0) | 1,715 (10.2) | 4,180 (16.7) | 3,243 (15.2) | |
| ≥Post-Secondary | 892 (10.9) | 890 (5.3) | 1,782 (7.1) | 1,590 (7.4) | |
| **Wealth quintile** | | | | | < .001 |
| Lowest | 1,758 (21.4) | 5,167 (30.7) | 8,734 (26.5) | 5,801 (27.1) | |
| Second | 1,349 (16.4) | 3,365 (20.0) | 7,104 (21.6) | 4,657 (21.8) | |
| Middle | 1,486 (18.1) | 2,961 (17.6) | 6,183 (18.8) | 4,037 (18.9) | |
| Fourth | 1,727 (21.0) | 2,904 (17.2) | 5,783 (17.6) | 3,703 (12.3) | |
| Highest | 1,889 (23.0) | 2,456 (14.6) | 5,144 (15.6) | 3,199 (15.0) | |
| **Place of residence** | | | | | < .001 |
| Urban | 4,701 (57.3) | 7,843 (46.5) | 12,544 (50.1) | 10,605 (49.6) | |
| Rural | 3,508 (42.7) | 9,010 (53.5) | 12,518 (49.9) | 10,792 (50.4) | |
| **In a relationship** | | | | | < .001 |
| Yes | 1,297 (15.8) | 13,755 (81.6) | 15,052 (60.1) | 15,048 (70.3) | |
| No | 6,912 (84.2) | 3,098 (18.4) | 10,010 (39.9) | 6,349 (29.7) | |
| **Number of living children** | | | | | < .001 |
| 0 | – | – | 8,209 (32.8) | 4,544 (21.2) | |
| 1 | – | – | 3,951 (15.8) | 3,951 (18.5) | |
| 2 | – | – | 3,456 (13.8) | 3,456 (16.2) | |
| 3 | – | – | 3,021 (12.1) | 3,021 (14.1) | |
| 4 | – | – | 2,474 (9.9) | 2,474 (11.6) | |
| 5 | – | – | 1,741 (7.0) | 1,741 (8.1) | |
| 6+ | – | – | 2,210 (8.8) | 2,210 (8.8) | |
| **Contraceptive method** | | | | | N/A |
| Not using | 7,150 (87.1) | 11,923 (70.7) | 19,073 (76.1) | 15,429 (72.1) | |
| Pill | 152 (1.9) | 652 (3.9) | 804 (3.2) | 804 (3.8) | |
| Injection | 123 (1.5) | 1,662 (9.9) | 1,785 (7.1) | 1,783 (8.3) | |
| Implant | 56 (0.8) | 1,261 (7.5) | 1,317 (5.3) | 1,316 (6.2) | |
| IUD | 3 (.04) | 99 (.59) | 102 (.41) | 102 (.48) | |
| Condom | 236 (2.9) | 148 (0.9) | 384 (1.5) | 381 (1.8) | |
| Periodic abstinence | 245 (3.0) | 538 (3.2) | 783 (3.1) | 774 (3.6) | |
| Withdrawal | 78 (1.0) | 143 (0.9) | 221 (0.9) | 220 (1.0) | |
| Emergency contraception | 142 (1.7) | 115 (0.7) | 257 (1.0) | 257 (1.2) | |
| Lactational amenorrhea method | 0 | 876 (5.2) | 876 (5.2) | 876 (5.2) | |
| Sterilization | 4 (0.1) | 223 (1.3) | 227 (.91) | 227 (1.1) | |
| Other method | 20 (4.8) | 89 (0.5) | 109 (.43) | 104 (.49) | |

*(Continued)*

**Table 1.** (Continued)

| Variable | 0 children N = 8,209 | 1+ children N = 16,853 | Full Sample N = 25,062 | Regression Sample N = 21,397 | P-value[a] |
|---|---|---|---|---|---|
| **Ever had sex** | | | | | < .001 |
| Yes | 4,544 (55.4) | 16,853 (100) | 21,397 (85.4) | 21,397 (100) | |
| No | 3,665 (44.6) | 0 | 3,665 (14.6) | 0 | |
| **Age at first sex, years** | | | | | < .001 |
| <15 | 415 (5.1) | 2,260 (13.4) | 2,675 (10.7) | 2,675 (12.5) | |
| 15–16 | 1,091 (13.3) | 4,810 (28.5) | 5,901 (23.6) | 5,901 (27.6) | |
| 17–18 | 1,386 (16.9) | 5,170 (30.7) | 6,556 (26.2) | 6,556 (30.6) | |
| 19–20 | 952 (11.6) | 2,979 (17.7) | 3,931 (15.7) | 3,931 (18.4) | |
| 21+ | 4,365 (53.2) | 1,634 (9.7) | 5,999 (23.9) | 5,999 (10.9) | |
| **Ever had an abortion** | | | | | < .001 |
| Yes | 649 (7.9) | 3,053 (18.1) | 3,702 (14.8) | 3,702 (14.8) | |
| No | 7,560 (92.1) | 13,800 (81.9) | 21,360 (85.2) | 21,360 (85.2) | |
| **Time since last intercourse** | | | | | < .001 |
| Within one week | 1,052 (23.2) | 5,307 (31.5) | 6,359 (29.7) | 6,359 (29.7) | |
| Within 1–4 weeks | 781 (17.2) | 3,801 (22.6) | 4,582 (21.4) | 4,582 (21.4) | |
| Within 1–12 months | 1,850 (40.7) | 5,425 (32.2) | 7,275 (34.0) | 7,275 (34.0) | |
| More than one year | 861 (18.9) | 2,320 (13.8) | 3,181 (14.9) | 3,181 (14.9) | |

Data presented as n (%).

[a]P-value of the variable against the outcome variable (use of highly effective contraception).

## Discussion

The analysis in the present study disaggregates the method of contraception used by Ghanaian women by the number of living children they have. Of particular interest to this analysis, number of living children, age, and timing of last intercourse were the most significant predictors of using modern contraception. Specifically, participants with <3 living children were less likely to use highly effective contraception (19.7%, 12.0%, and 5.6% for 0, 1, and 2 children respectively), while those with ≥4 living children were more likely (4.5%, 8.3%, and 11.4% for 4, 5, and 6+ children respectively). Participants older than age 40 and those who have not had sexual intercourse in "years" were 7.9% and 15.6% less likely to be using highly effective contraception. Among only women who were using contraception, having no living children was associated with a 22.3% reduced likelihood of using a highly effective method. Among those women, being in the lowest wealth quintile was associated with a 4.1% increased likelihood of using a highly effective method, while being wealthier was associated with a decreased likelihood (5.1% and 7.4% respectively for the fourth and fifth wealth quintiles). Women in rural areas and those who have had sex within one week are more likely to use highly effective contraception if they are using a method.

Previous research on this topic has had contradictory outcomes. Our findings are contrary to a study conducted in the Democratic Republic of Congo that found parity not significantly associated with the use of modern contraception, and a study in Botswana [21] that found a negative association between parity and modern contraceptive use. However, our findings support a study in Zambia [22] and one in Malawi [23], where higher parity was associated with contraceptive use. However, the study from Zambia showed that women who ever had a pregnancy and those with a history of abortion were less likely to be using contraception, while our analysis showed the opposite: women who have had an abortion are 4.5% more likely to be using highly effective contraception

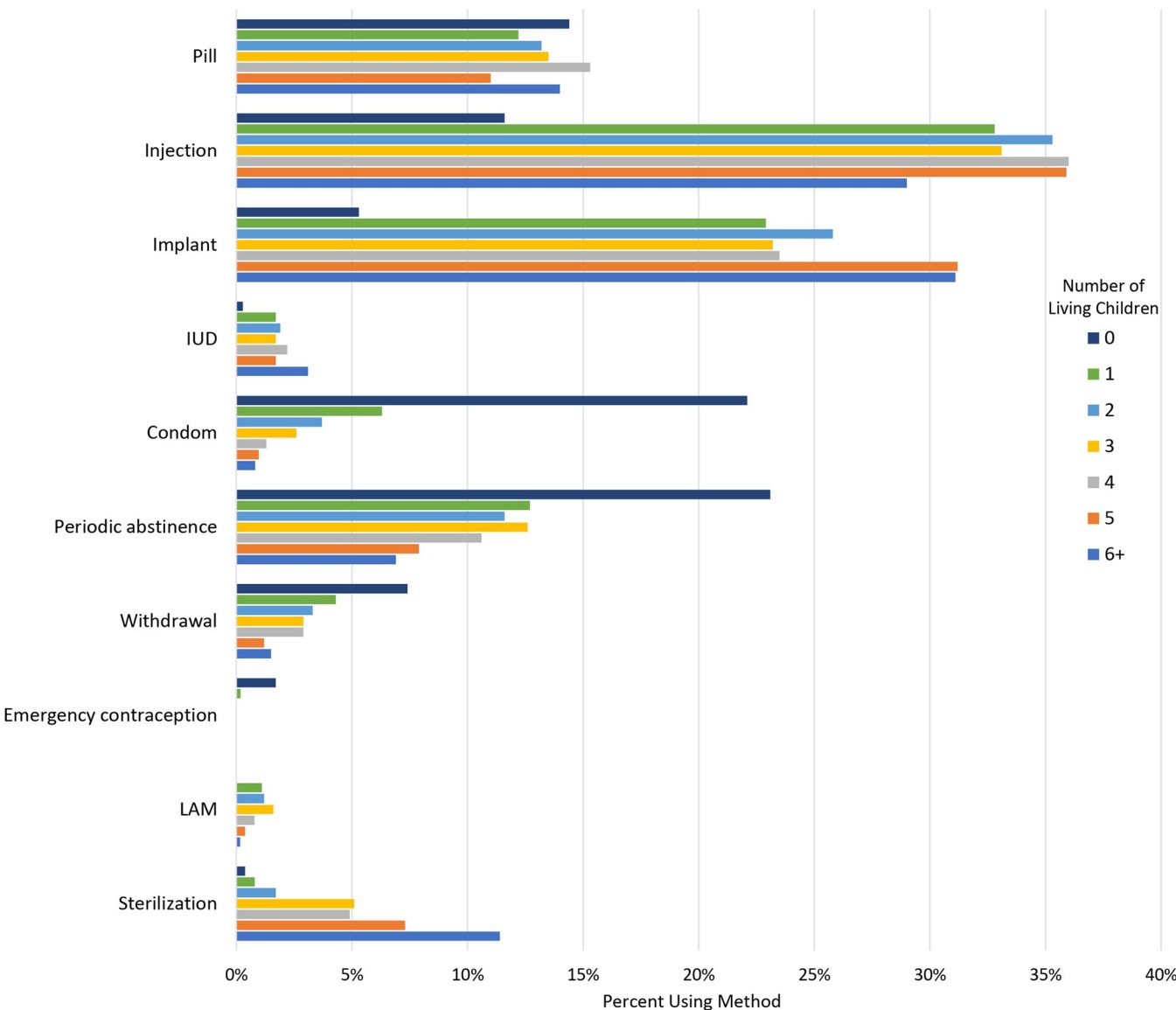

**Fig 1. Method of contraception used by number of living children.** LAM = lactation amenorrhea method; IUD = intrauterine device.

Our findings are consistent with research conducted in Accra, Ghana [24], which found women with lower levels of education being less likely to use modern contraception. Further, we found that higher education was associated with an increased likelihood of using highly effective contraception, as has been consistently found in previous work in Ghana [25,26] and across SSA [27–29].

This study also shows that older persons of reproductive age are more likely to use highly effective contraception. The result is similar to research in Malawi [23] and Ethiopia [30], where older age was negatively associated with using contraception. Previous research [23,25,31] has demonstrated that increasing wealth is associated with higher odds of using contraception. Our analysis indicates a reduced likelihood of using highly effective contraception for women in the highest wealth quintile, and no association at other levels of wealth. In Ghana, the most commonly used contraceptives (implant and injection), as well as most other

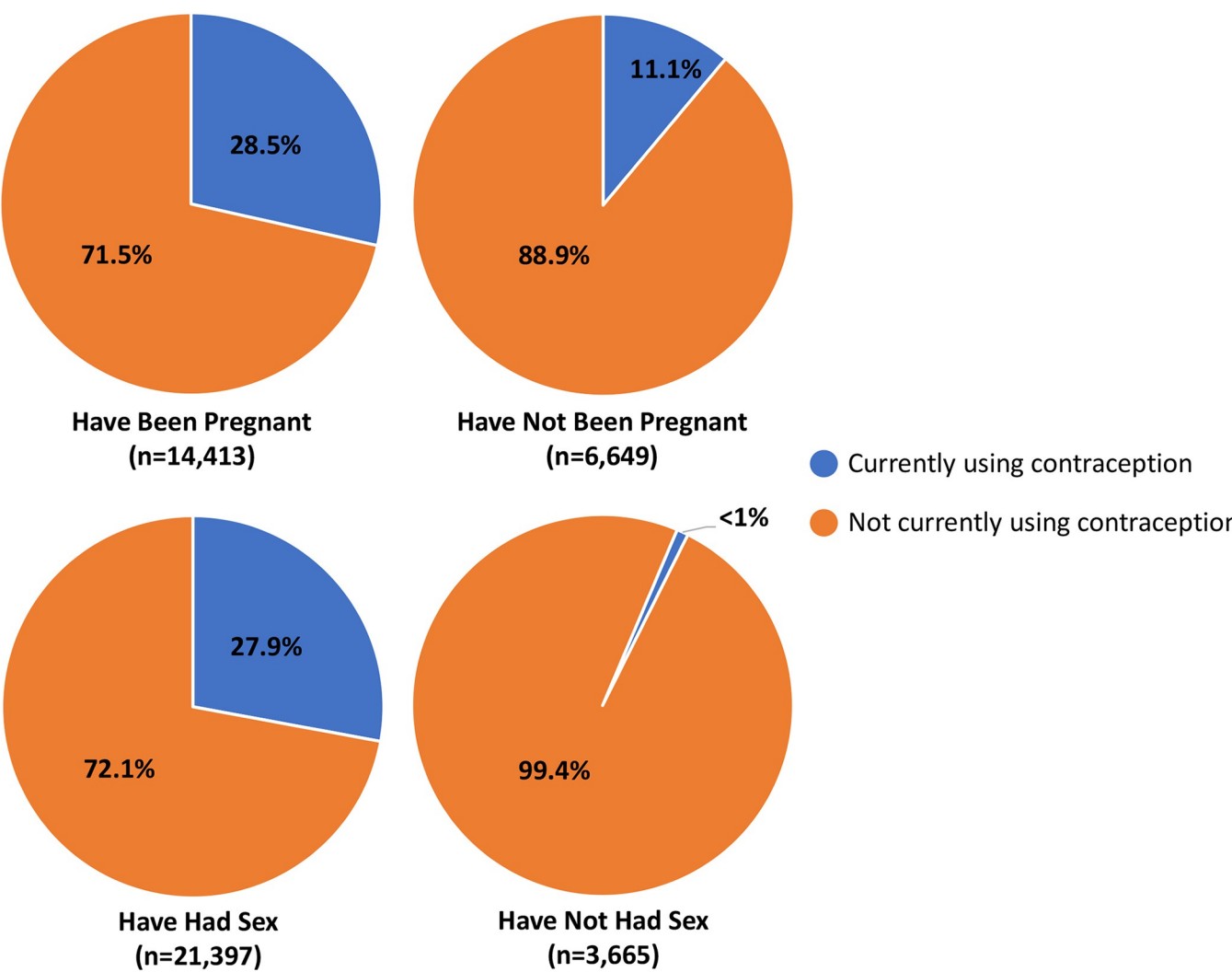

**Fig 2. Rates of current contraception use by pregnancy and sex history.**

methods, are available free of charge at many places, so it is not surprising that wealth is not associated with being able to obtain contraceptives.

Similar to previous work conducted in Nigeria [32], the vast majority of our participants are not using any modern method of contraception. However, unlike the study from Nigeria, our analysis shows an increasing likelihood of using modern contraception as parity increases. Perhaps the positive association between parity and modern contraceptive use could be explained from the point that nulliparous women and those with lower parity may be afraid of triggering primary or secondary infertility due to their reproductive choices, including using modern contraceptives [33]. However, in the case of multiparous and grand multiparous women, this fear may be non-existent, which may inform their higher likelihood of using modern contraceptives. Our findings are also contradictory to the study from Nigeria in terms of maternal age; our participants were less likely to be using highly effective contraception as age increased. In general, however, most of the women in this survey are not using any method of contraception, despite a strong family planning program and overall decreases in total fertility in the country. Qualitative work in Accra [34] has shown that many women use "periodic

**Table 2. Logistic regression illustrating factors associated with use of highly effective contraception[a,b].**

| Variable | Coefficient | 95% CI | p-value | Marginal Effect |
|---|---|---|---|---|
| **Age** | | | | |
| <19 | .204 | -.029- .438 | .087 | .032 |
| 19–24 | .585 | .458-.712 | < .001 | .098 |
| 25–29 | .324 | .217-.432 | < .001 | .052 |
| 30–34 | Ref | | | |
| 35–39 | -.212 | -.338- -.095 | < .001 | -.030 |
| 40+ | -.631 | -.759- -.504 | < .001 | -.079 |
| **Education** | | | | |
| None | -.330 | -.444- -.217 | < .001 | -.046 |
| Primary | Ref | | | |
| Junior Secondary | .194 | .091-.298 | < .001 | .031 |
| Senior Secondary | .158 | .025-.291 | .020 | .025 |
| ≥Post-Secondary | .204 | .019-.389 | .031 | .033 |
| **Wealth quintile** | | | | |
| Lowest | .025 | -.099-.148 | .697 | .004 |
| Second | .056 | -.058-.169 | .335 | .008 |
| Middle | Ref | | | |
| Fourth | .012 | -.105-.127 | .847 | .002 |
| Highest | -.120 | -.253- -.012 | .075 | -.018 |
| **Rural place of residence** | | | | |
| Rural | .156 | .057-.255 | .002 | .024 |
| **In a relationship** | -.281 | -.384- -.177 | < .001 | -.043 |
| **Number of living children** | | | | |
| 0 | -1.59 | -1.75- -1.42 | < .001 | -.197 |
| 1 | -.781 | -.912- -.649 | < .001 | -.120 |
| 2 | -.327 | -.453- -.202 | < .001 | -.056 |
| 3 | Ref | | | |
| 4 | .235 | .103-.367 | .001 | .045 |
| 5 | .422 | .271-.573 | < .001 | .083 |
| 6+ | .574 | .427-.720 | < .001 | .114 |
| **Age at first sex** | | | | |
| <15 | -.014 | -.133-.107 | .826 | -.002 |
| 15–16 | Ref | | | |
| 17–18 | .014 | -.079-.107 | .767 | .002 |
| 19–20 | .014 | -.089-.118 | .786 | .002 |
| 21+ | -.114 | -.259- .030 | .120 | -.017 |
| **Ever had an abortion** | .299 | .203-.394 | < .001 | .045 |
| **Time since last intercourse** | | | | |
| Within one week | .245 | .155-.334 | < .001 | .045 |
| Within 1–4 weeks | Ref | | | |
| Within 1–12 months | -.535 | -.630- -.440 | < .001 | -.082 |
| More than one year | -1.24 | -1.40- -1.09 | < .001 | -.156 |

[a]IUD, implant, injection, pill, or sterilization.

[b]The regression analysis was performed on complete cases, which was 21,397, as only those who answered they have had sex were asked the timing of their last sex.

**Table 3. Logistic regression showing factors associated with using highly effective contraception among the 5,968 women using contraception.**

| Variable | Coefficient | 95% CI | p-value | Marginal Effects |
|---|---|---|---|---|
| **Age** | | | | |
| <18 | -.286 | -.686-.115 | .162 | -.046 |
| 18–24 | .284 | .011-.558 | .042 | -.041 |
| 25–29 | .197 | -.034- .428 | .095 | .029 |
| 30–34 | Ref | | | |
| 35–39 | -.201 | -.436-.033 | .091 | -.032 |
| 40+ | -.437 | -.713- .162 | .002 | -.072 |
| **Education** | | | | |
| None | .328 | .067-.588 | .014 | .044 |
| Primary | Ref | | | |
| Junior Secondary | -.269 | -.464- -.074 | .007 | -.042 |
| Senior Secondary | -.664 | -.886- .442 | < .001 | -.112 |
| ≥Post-Secondary | -.655 | -.932- .377 | < .001 | -.110 |
| **Wealth quintile** | | | | |
| Lowest | .289 | .046-.531 | .020 | .041 |
| Second | .039 | -.169-.250 | .711 | .006 |
| Middle | Ref | | | |
| Fourth | -.314 | -.515- .112 | .002 | -.051 |
| Highest | -.448 | -.658- .237 | < .001 | -.074 |
| **Place of residence** | | | | |
| Rural | .475 | .307-.642 | < .001 | .073 |
| **In a relationship** | .128 | -.058-.315 | .178 | .020 |
| **Number of living children** | | | | |
| 0 | -1.16 | -1.47- .852 | < .001 | -.223 |
| 1 | -.213 | -.468-.041 | .100 | -.035 |
| 2 | .072 | -.172- .315 | .562 | .011 |
| 3 | Ref | | | |
| 4 | .257 | -.006-.520 | .056 | .037 |
| 5 | .542 | .198-.885 | .002 | .073 |
| 6+ | .506 | .161-.852 | .004 | .069 |
| **Age at first sex, years** | | | < .001 | |
| <15 | .063 | -.193-.318 | .630 | .009 |
| 15–16 | -.169 | -.339- .001 | .052 | -.026 |
| 17–18 | Ref | | | |
| 19–20 | -.211 | -.406- .015 | .034 | -.032 |
| 21+ | -.569 | -.800- .337 | < .001 | -.093 |
| **Ever had an abortion** | .029 | -.141- .120 | .736 | .005 |
| **Time since last intercourse** | | | | |
| Within one week | .232 | .059-.405 | .009 | .035 |
| Within 1–4 weeks | Ref | | | |
| Within 1–12 months | -.100 | -.275-.074 | .260 | -.016 |
| More than one year | -.122 | -.412-.168 | .410 | -.020 |

contraception," where they rely on periodic abstinence and using emergency contraception to avoid pregnancy if they have intercourse on days deemed unsafe, then rely on abortion if those methods fail. It is noted that these methods, especially fertility awareness methods, are under-counted in surveys such as Demographic and Health Surveys, which might explain our findings.

Women's choice of method changes with the number of living children they have. While periodic abstinence and condoms are the methods of choice for women with 0 living children, once they have children, women adopt injectable contraception and the implant at much higher rates—though still not at what would be considered 'high' rates. Sterilization, not surprisingly, becomes more common with higher numbers of living children. Previous work [34] has highlighted the importance of future fertility and regular menses for many women, which may explain why many women avoid hormonal contraception until after having one or more children.

While other studies [35] have found that women in relationships are more likely to use contraception, our findings show that women who say they are in a relationship—either married or living together—are 4.3% less likely to use a highly effective method of contraception. This supports more recent work in Ghana, which has shown the importance of natural family planning within relationships [34]. It is possible that women without living children want to use highly effective contraception but are limited by provider bias, where highly effective and long-term methods are only offered to parous women, as was true in a study in Tanzania [36].

The majority of participants in our study reported not using any method of contraception to avoid unwanted pregnancies. Future research that attempts to unpack the disconnect between not wanting to become pregnant and not using contraception is warranted.

This study is not without limitations. Using cross sectional data precludes our ability to make causal inferences. Further, the data are from 2017, and the situation may have changed since then. Due to the secondary nature of the data, we were limited in the analyses we were able to perform. Certain factors we would have controlled for, such as desired family size, were not in the survey, so we were not able to assess those. Finally, this analysis only investigated individual-level factors associated with the use of contraception. There are family and community factors not accounted for in this analysis that might be highly influential in a woman's decision to use or not use highly effective contraception.

## Conclusion

This study investigated the association between number of living children and contraceptive use. In our analysis, having a low level of education, being in a relationship, being older, having <3 living children, not having had sexual intercourse in months or years, and being in the highest wealth quintile are negatively associated with using highly effective contraception. Conversely, living in a rural area, having a history of abortion, having >3 living children, being younger, being more educated, and having had sex "within days" of the survey were positively associated with using highly effective contraception. However, more than eight in 10 survey respondents are not using any method of contraception. Understanding some of the seemingly contradictory findings, such as rural residence being negatively associated with using contraception, are deserving of further investigation.

## Author Contributions

**Conceptualization:** Sarah Compton, Emmanuel Nakua.

**Data curation:** Sarah Compton.

**Formal analysis:** Sarah Compton.

**Investigation:** Sarah Compton, Emmanuel Nakua, Cheryl Moyer, Veronica Dzomeku, Emily Treleaven, Easmon Otupiri, Jody Lori.

**Methodology:** Sarah Compton, Emmanuel Nakua, Cheryl Moyer, Veronica Dzomeku, Emily Treleaven, Easmon Otupiri, Jody Lori.

**Software:** Sarah Compton.

**Writing – original draft:** Sarah Compton, Emmanuel Nakua.

**Writing – review & editing:** Sarah Compton, Emmanuel Nakua, Cheryl Moyer, Veronica Dzomeku, Emily Treleaven, Easmon Otupiri, Jody Lori.

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
