## [Decision Letter · Decision Letter 0]

20 Sep 2023

PONE-D-23-19366Contraceptive use by number of living children in Ghana: evidence from the 2017 maternal health surveyPLOS ONE

Dear Dr. Compton,

Thank you for submitting your manuscript to PLOS ONE. After careful consideration, we feel that it has merit but does not fully meet PLOS ONE’s publication criteria as it currently stands. Therefore, we invite you to submit a revised version of the manuscript that addresses the points raised during the review process.Apparently there is an error regarding data availability since the MHS data is not available or conducted by PMA as indicated in the section on data availability. It is rather available from DHS repositories.The authors state that: ““What is the distribution of contraceptive use by parity in Ghana?” 99 This question remains unanswered in previous studies conducted in Ghana or other SSA countries 100 [18, 19]”. This is certainly an overstatement. Note that DHS data is regularly tabulated according to method used by parity. This is the case, for instance, in the MHS 2017 report, Table 4.13.1  Current use of contraception by background characteristics. This needs to be rephrased and the contribution of the paper made clear.PLOS ONE guidelines for statistical reporting are not being fulfilled. There is no information, for instance, on sample size or goodness of fit in table 2. Note also the discrepancies noted by reviewer 1.Regarding the logistic regression on highly effective contraception presented, what is the purpose on the analysis? It seems that women using highly effective contraception are compared to the rest of women, including those using less effective contraception. It seems that according to the stated goals, the interest is in finding how method choice changes with parity. This cannot be detected in this regression. You could either carry out a regression of effective vs less effective methods among users or a multinomial analysis. This discrepancy between stated goals and analysis hinders all the manuscript and needs to be addressed.Some of the terminology is not standard and should be clarified. Eg: Women “in a relationship”. Does this refer to women married on in a (cohabiting) union?There could be problems of collinearity among the variables included in the regression. For instance, including ever pregnant women together with parity and with having experience abortions.

We look forward to receiving your revised manuscript.

Kind regards,

José Antonio Ortega, Ph.D.

Academic Editor

PLOS ONE

Reviewers' comments:

Reviewer's Responses to Questions

**Comments to the Author**

1. Is the manuscript technically sound, and do the data support the conclusions?

Reviewer #1: Yes

Reviewer #2: Partly

2. Has the statistical analysis been performed appropriately and rigorously? 

Reviewer #1: No

Reviewer #2: Yes

3. Have the authors made all data underlying the findings in their manuscript fully available?

Reviewer #1: Yes

Reviewer #2: Yes

4. Is the manuscript presented in an intelligible fashion and written in standard English?

Reviewer #1: Yes

Reviewer #2: Yes

5. Review Comments to the Author

Reviewer #1: Date 15/08/2023

Dear editor,

Greetings

Thank you for asking me to review your paper titled “Contraceptive use by number of living children in Ghana: evidence from the 2017 maternal health survey”

General Concept

I think the paper is good but generally requires revision especially at the methods and results sections. It seeks to answer an important question of disaggregating parity and the use of contraceptives among women of childbearing age. Moreover, it can provide an important clue to the host country and policy makers on how contraceptives could be targeted among sexually active women by their number of living children. However, I have several concerns that need to be addressed. Here are a few comments to the authors.

Abstract

The strength of evidence using statistical significance (p values) must be indicated on page 3 lines 49-50. I recommend that the authors clearly define contraceptive use and coherently use this in all tables and figures.

Introduction

This part is very sound.

Methods

My significant concerns with the paper begin with the methods section followed by the results. The method section on statistical analysis does not indicate which method was used for the bivariate analysis such as Chi-square test of association between dichotomous variable and the demographic variables. Even though mentioned on page 7 line 138 that a bivariate analysis was performed, this result was never presented. The authors should include a bivariate analysis on table 2, as the results will help in understanding the final model selection criteria. Also, the final model presented using table 2 must be double checked as collinearity may exist between independent variables specifically (ever been pregnant, ever had an abortion and number of living children). After this check, the final model should be revised to exclude such variables for an accurate prediction of the adjusted model.

Results

This part needs a great improvement. All values in tables and figures must be checked.

The authors must carefully review and correct all errors on table 1 for the variables:

Ever had an abortion [3702+21360] ≠ regression sample total [21397]

Age at first sex years [2675+5901+6556+3931+5999] ≠ regression sample total [21397

Time since last intercourse [6359+4582+7275+3181] ≠ full sample total [25062]

Number presented on variable table 1 (Ever been pregnant [full model] [yes] =18413) and figure 2 (Have been pregnant, n= 14,413) are different and must be verified and corrected.

There are 8 categories (0,1..7+) for variable “Number of living children” presented on the two tables however the same variable has 7 categories (0,1..6+) presented at figure 1. The authors must ensure a consistent category is used throughout the manuscript.

Contraceptive method “LAM” is not shown on table 2 but presented at figure 1 whiles “Emergency contraceptives” is presented on table 2 but is not shown in figure 1.

I recommend that the authors consider including a mean/median age and standard deviation for the full sample and regression sample of respondents on contraceptive use on page 8 line 149. This will be useful in describing the demographic baseline characteristics of respondents. The updated table 2 should include an overall effect for each covariate as a supplement to the marginal/odds ratios provided for the bivariate and multivariate analysis.

Also ensure that you choose a consistent method of presenting ref/REF/Ref.

Discussion

The limitation section must be removed on page 16 line 261 as DHS data asks questions on fertility preferences, intentions, and desires- V602/5. Authors may add that the bias in self-reporting of total births may be misreported.

Conclusion

This part is relatively good though it can be more than this.

Reviewer #2: Authors should consider comparing the 2017 data with the current data in the 2022 Ghana Demographic and Health Survey and also the DHIMS 2 data from Ghana Health Service which will tell the real story. Hence comparative study is suggested.

6. PLOS authors have the option to publish the peer review history of their article (what does this mean?). If published, this will include your full peer review and any attached files.

Reviewer #1: **Yes: **Dennis Boateng

Reviewer #2: No

---

## [Author Response · Author response to Decision Letter 0]

16 Nov 2023

Editor’s review: 

• Apparently there is an error regarding data availability since the MHS data is not available or conducted by PMA as indicated in the section on data availability. It is rather available from DHS repositories.

o Thank you for this comment. We have updated the link in the data availability section of our submission to accurately represent where the data were obtained. 

• The authors state that: ““What is the distribution of contraceptive use by parity in Ghana?” 99 This question remains unanswered in previous studies conducted in Ghana or other SSA countries 100 [18, 19]”. This is certainly an overstatement. Note that DHS data is regularly tabulated according to method used by parity. This is the case, for instance, in the MHS 2017 report, Table 4.13.1 Current use of contraception by background characteristics. This needs to be rephrased and the contribution of the paper made clear.

o Thank you for this comment. We meant that it has not been reported in the published academic literature, but that was not clear the way it was written. We have made this adjustment. This leads into the larger question below (how does our analysis answer the stated question). We have thus updated the introduction to better represent both the question we are asking and how we sought to fill the gap in the literature.

• PLOS ONE guidelines for statistical reporting are not being fulfilled. There is no information, for instance, on sample size or goodness of fit in table 2. Note also the discrepancies noted by reviewer 1.

o Thank you for this. We have added information about the goodness-of-fit, as well as repeated the sample size for the regression models in Table 2 (it was already noted in Table 1 as the “Regression Sample”).

• Regarding the logistic regression on highly effective contraception presented, what is the purpose on the analysis? It seems that women using highly effective contraception are compared to the rest of women, including those using less effective contraception. It seems that according to the stated goals, the interest is in finding how method choice changes with parity. This cannot be detected in this regression. You could either carry out a regression of effective vs less effective methods among users or a multinomial analysis. This discrepancy between stated goals and analysis hinders all the manuscript and needs to be addressed.

o Response: This is a very good point. We have addressed this in a couple of ways:

We built a new model among those women using contraception, highly effective vs. less effective methods (which are condoms, periodic abstinence, withdrawal, and emergency contraception) to investigate the relationship between number of living children and which method those who are using a method choose. Among women using contraception, those with no living children are less likely to be using a highly effective method. 

We have also reframed the question to lead to the main model in the original version of the paper. We demonstrate earlier in the paper that, among those who use contraception, women with more living children differentially use methods. We did not want to build an ordered logit or probit model due to the fact that methods of contraception are not inherently ordered, and because of the complexity in interpreting the outcome. Thus, we have added the new model (described above) and made changes to the framing of the research question we sought to answer.

• Some of the terminology is not standard and should be clarified. Eg: Women “in a relationship”. Does this refer to women married on in a (cohabiting) union?

o Thank you for this comment. We have updated the methods to make it clear that “in a relationship” refers to women who are married, as well as those who have reported they are in a cohabitating union.

• There could be problems of collinearity among the variables included in the regression. For instance, including ever pregnant women together with parity and with having experience abortions.

o This point is well taken, as it was also raised by Reviewer 2. We have removed the “ever pregnant” variable, as it was indeed collinear with “ever sex” and “ever having an abortion.” Further, it was not adding to the model, so it has been removed. 

Reviewers’ comments:

Reviewer 1

General Concept

I think the paper is good but generally requires revision especially at the methods and results sections. It seeks to answer an important question of disaggregating parity and the use of contraceptives among women of childbearing age. Moreover, it can provide an important clue to the host country and policy makers on how contraceptives could be targeted among sexually active women by their number of living children. However, I have several concerns that need to be addressed. Here are a few comments to the authors.

Abstract

The strength of evidence using statistical significance (p values) must be indicated on page 3 lines 49-50. 

Response: Thank you. We have added this information to the results section of the abstract (all variables mentioned have a p-value of <.001). 

I recommend that the authors clearly define contraceptive use and coherently use this in all tables and figures.

Response: We have tried to define highly effective contraceptive use more clearly, including adding the definition in the background to more effectively set up our later analyses. 

Introduction

This part is very sound.

Methods

My significant concerns with the paper begin with the methods section followed by the results. The method section on statistical analysis does not indicate which method was used for the bivariate analysis such as Chi-square test of association between dichotomous variable and the demographic variables. Even though mentioned on page 7 line 138 that a bivariate analysis was performed, this result was never presented. The authors should include a bivariate analysis on table 2, as the results will help in understanding the final model selection criteria. 

Response: We have added that we conducted Chi-square analysis for the bivariate analysis and have added those associations in a p-value column added to Table 1. 

Also, the final model presented using table 2 must be double checked as collinearity may exist between independent variables specifically (ever been pregnant, ever had an abortion and number of living children). After this check, the final model should be revised to exclude such variables for an accurate prediction of the adjusted model. 

Thank you. This is an excellent point that was also noted by Reviewer 1. We have removed the “ever pregnant” variable, as it is collinear to the other variables and does not add to the model. 

Results

This part needs a great improvement. All values in tables and figures must be checked. 

The authors must carefully review and correct all errors on table 1 for the variables:

Ever had an abortion [3702+21360] ≠ regression sample total [21397] 

Age at first sex years [2675+5901+6556+3931+5999] ≠ regression sample total [21397 

Time since last intercourse [6359+4582+7275+3181] ≠ full sample total [25062] 

Response: The question, “how long ago did you last have sex” was only asked of those individuals who have ever had sex. Therefore, the regression analysis is restricted to those 21,397 women who reported they have had sex. However, the rest of the questions were asked of everyone, regardless of whether or not they have had sex. We feel it is important to include descriptives of all women, as a woman intending to use hormonal contraception, for example, should start that method before having sex. Moreover, we felt women who have not had sex is an important population to know more about. We have added that information to the text of the paper, as well as a footnote to Table 2. 

Number presented on variable table 1 (Ever been pregnant [full model] [yes] =18413) and figure 2 (Have been pregnant, n= 14,413) are different and must be verified and corrected.

Response: Due to collinearity, we have removed the “ever pregnant” variable from the regression. 

There are 8 categories (0,1..7+) for variable “Number of living children” presented on the two tables however the same variable has 7 categories (0,1..6+) presented at figure 1. The authors must ensure a consistent category is used throughout the manuscript.

Response: Thank you for this. We have combined these categories so they are now consistently reported as “6+ children.” 

Contraceptive method “LAM” is not shown on table 2 but presented at figure 1 whiles “Emergency contraceptives” is presented on table 2 but is not shown in figure 1.

Response: We had initially combined LAM into the “Other” category for the table, but have now added it as its own row. We have added emergency contraception to Figure 1. 

I recommend that the authors consider including a mean/median age and standard deviation for the full sample and regression sample of respondents on contraceptive use on page 8 line 149. This will be useful in describing the demographic baseline characteristics of respondents. The updated table 2 should include an overall effect for each covariate as a supplement to the marginal/odds ratios provided for the bivariate and multivariate analysis.

Also ensure that you choose a consistent method of presenting ref/REF/Ref.

Response: I’m afraid we do not know what you mean by “overall effect for each covariate.” Could the reviewer or editor clarify so that we can address this point? 

Discussion

The limitation section must be removed on page 16 line 261 as DHS data asks questions on fertility preferences, intentions, and desires- V602/5. Authors may add that the bias in self-reporting of total births may be misreported. 

Response: While the DHS reports on fertility preference, we have not found this variable in the Maternal Health Survey, which does not include the full set of questions included in the DHS. If we have missed this, we are happy to include that variable in the model, as we feel it is an important piece of using or not using contraception and which method a woman would choose to use. 

Conclusion

This part is relatively good though it can be more than this.

Response: Thank you for this comment. 

Reviewer 2:

I suggest the authors consider doing a comparative analysis by using the current data from the Ghana Demographic and Health Survey 2022, and also the DHIMS data from Ghana Health Service which has all the variables . I feel using only the 2017 data which is more than 5 years will not tell the real story. I suggest the authors consider during a comparative study by making use of current data from the GDHS and DHIMS data which are available at Ghana Service. These are all national data and are available.

Response: Thank you for this suggestion. However, the 2022 DHS data is not yet publicly available, so we are not able to access it via the DHS website. We had originally planned to use that dataset, but could not find the data available. While the preliminary results and key indicators are available, it appears the cleaning of the data are still underway, so the complete data are not currently available as far as we can tell. 

Consider showing a waiver from Ghana Health Service Ethics Committee 

Response: As these are publicly available data, we did not feel it necessary to have the study be approved by the Ghana Health Service Ethics Committee. 

This work was done in 2003, it seems too old to compare to the current study, please consider revising. 

Response: We agree that 2003 is a long time ago to compare to; however, there has not been much work in this space, so we feel it is important to use the literature that exists.

---

## [Decision Letter · Decision Letter 1]

29 Nov 2023

Contraceptive use by number of living children in Ghana: evidence from the 2017 maternal health survey

PONE-D-23-19366R1

Dear Dr. Compton,

We’re pleased to inform you that your manuscript has been judged scientifically suitable for publication and will be formally accepted for publication once it meets all outstanding technical requirements.

Kind regards,

José Antonio Ortega, Ph.D.

Academic Editor

PLOS ONE

Additional Editor Comments (optional):

The two reviewers and the editor feel that the major obstacles to publication have all been addressed.

Reviewers' comments:

Reviewer's Responses to Questions

**Comments to the Author**

1. If the authors have adequately addressed your comments raised in a previous round of review and you feel that this manuscript is now acceptable for publication, you may indicate that here to bypass the “Comments to the Author” section, enter your conflict of interest statement in the “Confidential to Editor” section, and submit your "Accept" recommendation.

Reviewer #1: All comments have been addressed

Reviewer #2: All comments have been addressed

2. Is the manuscript technically sound, and do the data support the conclusions?

Reviewer #1: Yes

Reviewer #2: Yes

3. Has the statistical analysis been performed appropriately and rigorously? 

Reviewer #1: Yes

Reviewer #2: Yes

4. Have the authors made all data underlying the findings in their manuscript fully available?

Reviewer #1: Yes

Reviewer #2: Yes

5. Is the manuscript presented in an intelligible fashion and written in standard English?

Reviewer #1: Yes

Reviewer #2: Yes

6. Review Comments to the Author

Reviewer #1: Thank you very much for giving me the opportunity to review the paper again. I have gone through the latest version and many sections has been improved especially in the methods and results section.

The current version has addressed the issues in the methodology of collinearity and inconsistencies previously found in the results.

Reviewer #2: The authors have addressed all comments, and the manuscript is fit for publication. I have no further comments.

7. PLOS authors have the option to publish the peer review history of their article (what does this mean?). If published, this will include your full peer review and any attached files.

Reviewer #1: **Yes: **Dennis Boateng

Reviewer #2: No

---

## [Editor Report · Acceptance letter]

6 Dec 2023

PONE-D-23-19366R1 

Contraceptive use by number of living children in Ghana: evidence from the 2017 maternal health survey 

Dear Dr. Compton:

I'm pleased to inform you that your manuscript has been deemed suitable for publication in PLOS ONE. Congratulations! Your manuscript is now with our production department. 

Kind regards, 

on behalf of

Dr. José Antonio Ortega 

Academic Editor

PLOS ONE